# VL-JEPA: Joint Embedding Predictive Architecture for Vision-language

**Delong Chen**[1,2,*], **Mustafa Shukor**[1,3,*], **Théo Moutakanni**[1,*], **Willy Chung**[1,3,*],
**Jade Yu**[1], **Tejaswi Kasarla**[1,5], **Allen Bolourchi**[1,6], **Yann LeCun**[1,4], **Pascale Fung**[1,2]

[1]Meta FAIR, [2]HKUST, [3]Sorbonne Université, [4]NYU, [5]University of Amsterdam, [6]USC
[*]Equal contribution
delong.chen@connect.ust.hk

## Abstract

We introduce VL-JEPA, a vision-language model built on a Joint Embedding Predictive Architecture (JEPA). Instead of autoregressively generating tokens as in classical VLMs, VL-JEPA predicts *continuous embeddings* of the target texts. By learning in an abstract representation space, the model focuses on task-relevant semantics while abstracting away surface-level linguistic variability. In a strictly controlled comparison against standard token-space VLM training with the same vision encoder and training data, VL-JEPA achieves stronger performance while having 50% fewer trainable parameters. At inference time, a lightweight text decoder is invoked only when needed to translate VL-JEPA predicted embeddings into text. We show that VL-JEPA natively supports *selective decoding* that reduces the number of decoding operations by $\sim 2.85\times$ while maintaining similar performance compared to non-adaptive uniform decoding. Beyond generation, the VL-JEPA's embedding space naturally supports open-vocabulary classification, text-to-video retrieval, and discriminative VQA *without any architecture modification*. On eight video classification and eight video retrieval datasets, the average performance VL-JEPA surpasses that of CLIP, SigLIP2, and Perception Encoder. At the same time, the model achieves comparable performance as classical VLMs (InstructBLIP, QwenVL) on four VQA datasets: GQA, TallyQA, POPE and POPEv2, despite only having 1.6B parameters.

## 1 Introduction

One of the most important aspects of advanced machine intelligence is the ability to understand the physical world that surrounds us. This ability enables AI systems to learn, reason, plan and act in the real world in order to assist humans (LeCun, 2022). Intelligent systems that need to act in the real world includes wearable devices and robots (Fung et al., 2025). Machine learning tasks that make up for this ability include captioning, retrieval, visual question answering, action tracking, reasoning and planning etc (Bordes et al., 2024; Chen et al., 2025b). Systems for such real-world applications must have real-time response with low latency and inference cost.

Currently, the common approach to achieve these tasks is to use large token-generative Vision Language Models (VLMs) (Liu et al., 2023; Dai et al., 2023; Alayrac et al., 2022; Chen et al., 2024b; Cho et al., 2025; Chen et al., 2022), which takes visual input $X_V$, textual query $X_Q$ to generate desired textual response $Y$ autoregressively in token space, *i.e.*, $(X_V, X_Q) \mapsto Y$. This is straightforward but inadequate for two main reasons. First, VLMs are expensive to develop, because they are trained to generate responses $Y$ to queries by capturing both task-relevant semantics with task-irrelevant surface linguistic features such as words choice, style or paraphrasing. During training, VLMs must model both aspects, which results in unnecessary computing effort spent producing diverse token sequences that ultimately do not impact the correctness of the output. Second, real-time tasks involving live streaming video (*e.g.,* live action tracking) require sparse and selective decoding (*e.g.,*, emitting a description only when a new event occurs) (Zhou et al., 2024). However, VLMs rely on autoregressive token-by-token decoding, which must be completed before revealing the un-

derlying semantics of $Y$. This process introduces unnecessary latency and hampers the ability to update semantics dynamically in real time.

This paper introduces the Joint Embedding Predictive Architecture for Vision-Language (VL-JEPA), turning expensive learning of data-space token generation into more efficient latent-space semantic prediction. As illustrated in Fig. 1, the model employs **x-encoder** to map vision inputs $X_V$ into embedding $S_V$, a **y-encoder** to map the textual target $Y$ into an embedding $S_Y$, and a **predictor** that learns the mapping $(S_V, X_Q) \mapsto S_Y$ where $X_Q$ is a textual query (*i.e.,* the prompt). The training objective is defined in the embedding space $\mathcal{L}_{\text{VL-JEPA}} = D(\hat{S}_Y, S_Y)$ instead of the data space $\mathcal{L}_{\text{VLM}} = D(\hat{Y}, Y)$. During inference, a **y-decoder** reads out the predicted embedding $\hat{S}_Y$ to text space $\hat{Y}$ when needed.

Thanks to its **non-generative** nature, VL-JEPA is not forced to reconstruct every surface detail of $Y$ in the token space. Instead, it only needs to predict the abstract representation $S_Y$ in the embedding space. In the raw one-hot token space, different plausible $Y$ outputs for the same input often appear nearly orthogonal if they don't share overlapping tokens. However, in the embedding space, these diverse targets can be mapped to nearby points that share similar semantics. This simplifies the target distribution thus makes the learning process more efficient. In addition, unlike VLMs, this approach eliminates the need for learning language generation with a heavy decoder during training, resulting in significant efficiency gains.

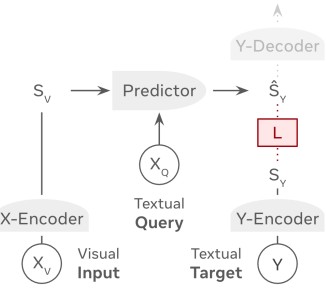

Figure 1: **VL-JEPA.**

Thanks to its **non-autoregressive** nature, VL-JEPA can produce continuous streams of target semantic embeddings within sliding windows with minimal latency as it only require a single forward pass without autoregressive decoding. This is particularly advantageous for real-time online applications such as live action tracking, scene recognition, or planning, where the embedding stream can be selectively decoded by a lightweight y-decoder, enabling efficient and prompt updates.

In this work, we empirically validate the advantages of VL-JEPA. We conduct a strictly controlled comparison against classical token-generative VLM (Liu et al., 2023; Cho et al., 2025): both setups use the same vision encoder, spatial resolution, frame rate, training data, batch size, and number of iterations, etc., with the *only* difference being the objective in token space or embedding space. Under this matched training condition, VL-JEPA delivers consistently higher performance on zero-shot captioning and classification while using roughly half the trainable parameters, indicating that embedding-space supervision improves learning efficiency.

Beyond the training phase, VL-JEPA also delivers substantial inference-time efficiency improvement through *selective decoding*, where decoding happens only due to significant change in the predicted embedding stream. Empirically, this strategy reduces the number of decoding operations by $\sim 2.85\times$ while preserving overall output quality measured by average CIDEr scores.

Our final VL-JEPA models are trained in two stages: 1) a pretraining stage using caption data to establish robust vision-language alignment, and 2) a supervised finetuning (SFT) stage that equips the model with VQA capabilities. The model resulting from the first stage, denoted as **VL-JEPA$_{\text{BASE}}$**, is evaluated on *zero-shot* classification and text-to-video retrieval. VL-JEPA$_{\text{BASE}}$ outperforms CLIP (Radford et al., 2021), SigLIP2 (Tschannen et al., 2025), and Perception Encoder (Bolya et al., 2025) models in terms of average classification accuracy (across 8 datasets) and retrieval recall@1 (across 8 datasets). Following the second stage, the resulting **VL-JEPA$_{\text{SFT}}$** demonstrates significantly improved classification performance due to its exposure to in-domain training data. As a unified *generalist* model, VL-JEPA$_{\text{SFT}}$ approaches the performance of *specialist* models optimized for individual benchmarks. Simultaneously, VL-JEPA$_{\text{SFT}}$ exhibits effective VQA capabilities, achieving performance on par with established VLM families, such as InstructBLIP (Dai et al., 2023) and Qwen-VL (Bai et al., 2023), across four datasets covering compositional visual reasoning (Hudson & Manning, 2019), complex object counting (Acharya et al., 2019), and object hallucination (Li et al., 2023b; 2025b).

In summary, the contributions of this paper are as follows:

- We introduce VL-JEPA, the first non-generative model that can perform general-domain vision-language tasks in real-time, built on a joint embedding predictive architecture.

- We demonstrate in controlled experiments that VL-JEPA, trained with latent space embedding prediction, outperforms VLMs that rely on data space token prediction.

- We show that VL-JEPA delivers significant efficiency gains over VLMs for online video streaming applications, thanks to its non-autoregressive design and native support for selective decoding.

- We highlight that our VL-JEPA$_{\texttt{SFT}}$ model, with an unified model architecture, can effectively handle a wide range of classification, retrieval, and VQA tasks at the same time.

## 2 METHODOLOGY

We propose **VL-JEPA** (Fig. 1), a model with the joint embedding predictive architecture (JEPA) for vision-language tasks. VL-JEPA is trained with triplets $\langle X_V, X_Q, Y \rangle$, where $X_V$ denotes the **visual input** (a single image or a sequence of video frames), $X_Q$ is a **textual query** (*i.e.,* a question) and $Y$ is the **textual target** (*i.e.,* the answer) to be predicted. The VL-JEPA comprises of four components:

1. **X-Encoder** $(X_V \mapsto S_V)$ compresses high-volume visual inputs to compact visual embeddings–a sequence of continuous vectors analogous to "visual tokens" in classical VLMs.

2. **Predictor** $(\langle S_V, X_Q \rangle \mapsto \hat{S}_Y)$ is the core component of VL-JEPA. It maps visual embeddings to a prediction of target embedding, with a textual query as conditioning.

3. **Y-Encoder** $(Y \mapsto S_Y)$ embeds the textual target into a continuous latent space as the prediction target. The target embedding is expected to abstract away task irrelevant information.

4. **Y-Decoder** $(\hat{S}_Y \mapsto \hat{Y})$ is not involved during the main training phrase of VL-JEPA. At inference time, it translates the predicted embedding as human-readable text when necessary.

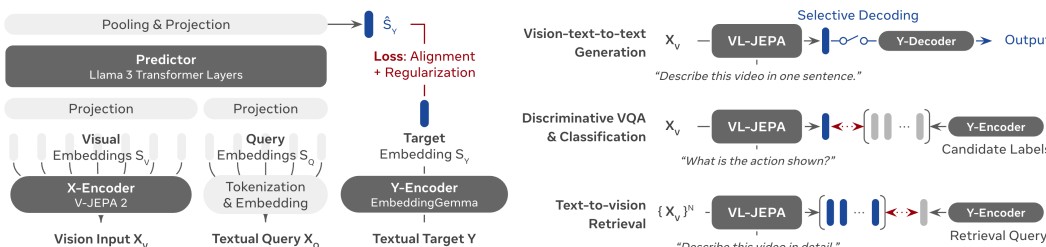

Figure 2: **Left: VL-JEPA Architecture**. It learns to predict the target embedding $S_Y$, instead of reconstructing the raw target $Y$ in token space as in classical VLMs. **Right: VL-JEPA Applications**: It handles vision-text-to-text generation tasks (*e.g.,* captioning) with selective decoding mechanism natively supported. Furthermore, VL-JEPA's embedding space facilitates discriminative VQA, open-vocabulary classification and text-to-video retrieval tasks using a single unified model architecture.

Fig. 2 illustrates how we instantiate the VL-JEPA architecture in this paper. For the `X-Encoder`, we chose V-JEPA 2 (Assran et al., 2025), a Vision Transformer that outputs a sequence of visual tokens, which are then projected and fed into the `Predictor` initialized using Llama 3 Transformer layers. Query conditioning is achieved by tokenizing and embedding the textual query and feeding the resulting textual token embeddings into the `Predictor` along with the visual embeddings. The outputs of the Llama 3 Transformer layers are pooled and projected into the target embedding space produced by the `Y-Encoder`, which is initialized by EmbeddingGemma-300M (Vera et al., 2025). We provide more technical details in §3.

**Training Objective.** JEPA models typically optimize two objectives jointly: 1) prediction error in the embedding space, and 2) additional regularization that avoids representation collapse (Bardes et al., 2021; Balestriero & LeCun, 2025). Any loss that implements these two properties can be

applied to VL-JEPA. Alternatively, the regularization term can be replaced by other anti-collapse strategies, such as using an exponential moving average (EMA) for the `Y-Encoder` (Assran et al., 2025) or freezing the `Y-Encoder` (Zhou et al., 2025).

In this work, we adopt the **InfoNCE loss** (Radford et al., 2021) due to its maturity in the vision-language domain. More advanced non-sample-contrastive regularization, such as VICReg (Bardes et al., 2021) and SIGReg (Balestriero & LeCun, 2025) can also be applied but we leave the exploration to future works. InfoNCE loss can be mathematically divided (Wang & Isola, 2020) into: 1) a representation *alignment* term that minimizes the distance between normalized prediction and target embeddings, and 2) a *uniformity* regularization term that pushes embeddings in a batch apart from each other, thus avoiding representation collapse. We train the `Predictor` and the `Y-Encoder` jointly with bi-directional InfoNCE loss, enabling them to mutually learn from each other.

Compared to the token-space loss used by generative VLMs, calculating the training loss in the embedding space is beneficial due to the **simplified target distribution**. Specifically, many real-world prediction tasks are inherently ill-posed: for the same input $X$, there may exist multiple plausible targets $Y$ that are all acceptable. For example, given the query *"What will happen here if I flip this light switch down?"*, both *"the lamp is turned off"* and *"room will go dark"* are valid answers. In the raw one-hot token space, however, the two sequences are orthogonal since they share no overlapping tokens. But when VL-JEPA's `Y-Encoder` embeds them into nearby points (ideally yielding a compact unimodal distribution), the learning task becomes much easier: the model no longer needs to fit multiple disjoint high-density regions in sparse token space, but only a single coherent mode in a continuous embedding space.

**Multi-tasking.** VL-JEPA supports diverse tasks using a *single*, *unified* architecture (Fig. 2). For vision-text-to-text generation tasks, such as captioning or open-ended VQA, the query $X_Q$ is a captioning prompt or a question, and the predictor learns to predict the embedding of the target output, $\hat{S}_Y$, which is then decoded into text. VL-JEPA also supports CLIP-style open-vocabulary classification and discriminative VQA, where candidate label texts are encoded into embeddings and compared with prediction $\hat{S}_Y$ to select the nearest match. For text-to-video retrieval, candidate videos are mapped to their predicted embeddings $\hat{S}_Y$ using a retrieval a captioning prompt, and then ranked by similarity to the encoded textual retrieval query.

**Selective Decoding.** Real-world video applications often require online streaming inference, such as tracking user actions in smart glasses for procedural assistance (Chen et al., 2024c), monitoring world states for online planning, navigation and robotics (Shukor et al., 2025; Black et al., 2025; Song et al., 2025). A central challenge is balancing two competing needs: the model must continuously update semantics as new frames arrive, but computational efficiency and latency are critical.

Existing VLMs typically rely on explicit memory mechanisms (Zhou et al., 2024; Qian et al., 2024) to decide when to decode or complex KV-cache optimizations (Di et al., 2025) for efficiency, since autoregressive language models are expensive to run continuously. VL-JEPA, in contrast, natively supports selective decoding. Since it predicts a semantic answer embedding non-autoregressively, the model provides a continuous semantic stream of $\hat{S}_Y$ that can be monitored in real time. This stream can be stabilized with simple smoothing (*e.g.,* average pooling) and decoded only when a significant semantic shift is detected, such as when the local window variance exceeds a threshold. In this way, VL-JEPA maintains always-on semantic monitoring while avoiding unnecessary decoding, achieving both responsiveness and efficiency.

## 3 IMPLEMENTATION OF VL-JEPA

### 3.1 MODEL ARCHITECTURE

**X-Encoder.** Unless otherwise specified, we use a frozen `V-JEPA 2 ViT-L` (Assran et al., 2025) with 304M parameters, a self-supervised vision model that excels at both image and video tasks. Each video input is uniformly sampled into frames at $256^2$ resolution. For image inputs, the same image is duplicated to match the input shape.

**Predictor.** The predictor is initialized with the last 8 Transformer layers of `Llama-3.2-1B`, resulting in 490M trainable parameters. The text tokenizer and token embedding are also from

`Llama-3.2-1B`. We allow maximum 512 query tokens, and put `[PAD]` tokens for short queries. We disable the causal attention mask so that both vision and query embeddings can be jointly attended. Linear projections connect the predictor with the vision and text embeddings, and average pooling on non-`[PAD]` tokens is applied to obtain the predicted target embedding.

**Y-Encoder.** We use `EmbeddingGemma-300M` (Vera et al., 2025) as the initialization of the `Y-Encoder`. We set maximum context length of 512 to handle detailed captions. We found that setting a learning rate multiplier of $\times 0.05$ to all text encoder parameters improves performance, since the quality of embedding prediction would be suboptimal in the beginning of training. Linear projection head is applied to both `Predictor` and `Y-Encoder`, obtaining a shared embedding space with 1,536 dimensions, where the loss is calculated.

## 3.2 Two-stage Training

**Large-scale Pretraining.** VL-JEPA is trained with two stages. The first query-free pretraining stage aims to establish robust vision-language alignment using massive caption data. We use Datacomp (Gadre et al., 2023) and YFCC-100M (Thomee et al., 2016) for image-text data. For video-text data, we use ACTION100M (Chen et al., 2026), which consists action description and video captions generated on HowTo100M videos (Chen et al., 2025b).

We first do image-only training on Datacomp and YFCC-100M with only 1 frame per visual input, which allows us to use a large batch size of 24k. After 100k iterations, the model has seen 2B samples and achieved 61.6% ImageNet zero-shot accuracy (without prompt ensembling). Then, we continue with video pretraining with 8 frames per input for 60k iterations and 32 frames for 10k iterations in the end. The pretraining takes 4 weeks using 24 nodes with $8\times$NVIDIA H200 GPUs each. We adopt a constant learning rate of $5\times10^{-5}$ to facilitate extended training. We refer the readers to the Action100M paper (Chen et al., 2026) for more details. We call the resulting model **VL-JEPA$_{\text{BASE}}$** and measure *zero-shot* classification and retrieval performance with this model.

**Supervised Finetuning.** The second query-conditioned supervised finetuning (SFT) stage empowers VL-JEPA VQA capabilities while maintaining the pretrained vision-language alignment for classification and retrieval. The training data is selected from the PLM data mixture (Cho et al., 2025), including 25M VQA samples, 2.8M captioning samples, 1.8M classification samples, and downsampled pretraining stage data to avoid catastrophic forgetting.

We train the model for 83k steps with a batch size of 3,072 ($\sim$2.5s days with 24 nodes), with cosine learning rate annealing applied to improve convergence. Since excessive human labeled data is included in this SFT data mixture, we no longer emphasize *zero-shot* evaluation for the resulting **VL-JEPA$_{\text{SFT}}$** from this stage. Instead, we evaluate VQA capabilities and compare it with state-of-the-art *specialist* models.

## 4 Experiments

We begin by evaluating VL-JEPA's classification and retrieval performance in §4.1, and benchmark VL-JEPA on VQA datasets in §4.2. We demonstrate application of VL-JEPA for understanding the relationship between world state changes and action concepts (*i.e.,* inverse dynamics) in §4.3. In §4.4, we demonstrate the advantage of embedding prediction by comparing it with a token-predictive VLM baseline under a strictly controlled setting. In §4.5, we evaluate the effectiveness of VL-JEPA's selective decoding, and show that it reduces decoding cost while maintaining the performance. Next, we analyze VL-JEPA's `Y-Encoder` in §4.6. Finally, we present ablation studies in §4.7.

## 4.1 Classification and Retrieval

**Evaluation Setup.** We evaluate VL-JEPA following the CLIP-style evaluation protocol (see Fig.2 and §2 "Multi-tasking"). We assess VL-JEPA on a broad suite of benchmarks, including 8 classification datasets and 8 retrieval datasets. For *zero-shot* evaluation, we compare against *generalist foundation models* CLIP (Radford et al., 2021), SigLIP2 (Tschannen et al., 2025), and Perception Encoder (PE-Core)(Bolya et al., 2025). We additionally report reference numbers from *specialist models* that are individually optimized for each benchmark.

Table 1: **Video classification and text-to-video retrieval**. Best *zero-shot* performance in each dataset are **highlighted**. Samples seen = training step × effective batch size.

| Model | | # Parameters | # Samples Seen | Zero-shot | Generalist Model | Video Classification (Top-1 Accuracy) | | | | | | | | | Text-to-video Retrieval (Recall@1) | | | | | | | | |
|---|---|---|---|---|---|---|---|---|---|---|---|---|---|---|---|---|---|---|---|---|---|---|---|
| | | | | | | Average | SSv2 (Goyal et al., 2017) | EK100 (Damen et al., 2022) | EgoExo4D (Grauman et al., 2024) | Kinetics-400 (Kay et al., 2017) | COIN (SR) (Tang et al., 2019) | COIN (TR) (Tang et al., 2019) | CrossTask (SR) (Zhukov et al., 2019) | CrossTask (TR) (Zhukov et al., 2019) | Average | MSR-VTT (Xu et al., 2016) | ActivityNet (Caba Heilbron et al., 2015) | DiDeMo (Anne Hendricks et al., 2017) | MSVD (Chen & Dolan, 2011) | YouCook2 (Zhou et al., 2018) | PVD-Bench (Bolya et al., 2025) | Dream-1k (Wang et al., 2024) | VDC-1k (Chai et al., 2024) |
| CLIP | RN50 | 75M | 12.8B | | | 21.8 | 2.1 | 1.5 | 2.1 | 41.4 | 8.6 | 39.0 | 10.9 | 68.7 | 28.3 | 28.7 | 17.7 | 24.7 | 29.7 | 5.1 | 27.6 | 47.2 | 46.0 |
| | ViT-B | 124M | 12.8B | ✓ | ✓ | 25.4 | 3.1 | 1.3 | 2.8 | 49.5 | 11.2 | 47.3 | 16.2 | 71.5 | 29.3 | 31.0 | 19.5 | 25.7 | 34.0 | 6.1 | 27.0 | 48.5 | 42.9 |
| | ViT-L | 389M | 12.8B | | | 30.7 | 3.8 | 3.7 | 2.6 | 58.3 | 14.7 | 63.5 | 20.8 | 78.5 | 35.3 | 35.9 | 23.4 | 30.7 | 41.9 | 7.9 | 36.7 | 56.8 | 49.3 |
| SigLIP2 | ViT-B | 375M | 40B | | | 33.9 | 5.2 | 2.3 | 4.5 | 57.8 | 20.6 | 69.9 | 27.7 | 82.9 | 39.6 | 40.2 | 25.0 | 32.1 | 48.6 | 13.8 | 52.1 | 60.9 | 43.7 |
| | ViT-L | 882M | 40B | ✓ | ✓ | 38.6 | 5.9 | 4.5 | 6.4 | 63.6 | 24.2 | 78.5 | 35.1 | 90.8 | 45.4 | 41.6 | 32.7 | 35.1 | 53.5 | 19.0 | 59.2 | 71.6 | 50.9 |
| | ViT-g | 1.9B | 40B | | | 39.8 | 6.1 | 6.1 | 5.6 | 68.0 | 26.0 | 80.4 | 35.1 | 90.8 | 47.5 | 43.4 | 33.9 | 38.9 | 56.0 | 22.2 | 60.4 | 73.0 | 52.5 |
| PE-Core | ViT-B | 448M | 58B | | | 37.2 | 5.8 | 3.3 | 6.0 | 65.4 | 21.5 | 77.1 | 26.9 | 91.8 | 44.9 | 46.5 | 35.4 | 35.3 | 49.1 | 15.2 | 59.8 | 68.7 | 49.2 |
| | ViT-L | 671M | 58B | ✓ | ✓ | 42.9 | 9.3 | 6.0 | 11.6 | 73.4 | 27.1 | 83.3 | 37.5 | 95.3 | 50.2 | 48.9 | 41.7 | 40.8 | 56.2 | 22.5 | 64.7 | 75.9 | 51.0 |
| | ViT-G | 2.3B | 86B | | | 44.7 | 9.0 | 6.4 | 13.6 | **76.4** | 29.0 | **86.0** | 40.3 | **97.2** | 58.1 | **51.6** | 49.1 | 44.5 | **58.7** | 26.0 | 77.0 | 89.2 | 68.5 |
| **VL-JEPA**BASE | ViT-L | 1.6B | 3.3B | ✓ | ✓ | **52.5** | **19.3** | **21.8** | **33.2** | 64.8 | **47.4** | 79.4 | **64.5** | 89.6 | **63.7** | 40.0 | **64.9** | **50.0** | 49.0 | **40.4** | **83.1** | **93.3** | **88.8** |
| **VL-JEPA**SFT | ViT-L | 1.6B | 3.6B | ✗ | ✓ | 75.4 | 73.2 | 44.6 | 68.1 | 84.8 | 66.4 | 90.3 | 79.8 | 96.2 | 63.8 | 46.2 | 62.4 | 52.3 | 52.3 | 37.8 | 82.6 | 89.3 | 87.7 |
| *Previous SoTA (including specialist models)* | | | | ✗ | ✗ | - | 77.5 | 56.4 | 47.8 | 92.1 | 67.3 | 95.3 | 64.5 | 96.0 | - | 62.8 | 74.1 | 74.2 | 61.4 | 28.9 | 77.0 | 89.2 | 68.5 |

Table 2: **VQA benchmarks.** We report accuracy on GQA (Hudson & Manning, 2019), TallyQA (Acharya et al., 2019), POPE (Li et al., 2023b), and POPEv2 (Li et al., 2025b). Scores lower than our model are marked in red. Scores from SmolVLM are obtained by our evaluation, while other baselines are reported in the literature.

| *GQA: compositional visual reasoning* | | | *TallyQA: complex object counting* | | | *POPE: object hallucination* | | | *POPEv2: object hallucination* | |
|---|---|---|---|---|---|---|---|---|---|---|
| **Model** | **Accuracy** | | **Model** | **Accuracy** | | **Model** | **Accuracy** | | **Model** | **Accuracy** |
| BLIP-2 (OPT-2.7B) | 33.9 | | SmolVLM-256M | 32.3 | | SmolVLM2-256M | 56.4 | | SmolVLM-256M | 62.3 |
| BLIP-2 (FlanT5XXL) | 41.0 | | SmolVLM-500M | 44.8 | | SmolVLM-256M | 57.9 | | LLaVA-1.5-13B | 72.7 |
| InstructBLIP (FlanT5XL) | 48.4 | | PaLI-700M | 62.3 | | LLaVA-7B | 72.9 | | InternVL2-8B | 74.5 |
| InstructBLIP (Vicuna-13B) | 49.5 | | SmolVLM-2B | 64.7 | | InstructBLIP (Vicuna-13B) | 79.0 | | InternVL2-26B | 76.1 |
| Qwen-VL-Chat-7B | 57.5 | | PaLI-3B | 65.8 | | Video-LLaVA (7B) | 83.4 | | Qwen2-VL-72B | 79.4 |
| Qwen-VL-7B | 59.3 | | InstructBLIP (Vicuna-13B) | 68.0 | | SmolVLM-500M | 85.8 | | SmolVLM-500M | 83.8 |
| InternVL-Chat (Vicuna-7B) | 59.5 | | PaLI-17B | 71.9 | | LLaVA-1.5-7B | 85.9 | | Qwen2-VL-7B | 87.0 |
| LLaVA-1.5 (Vicuna-7B) | 62.0 | | LLaVA-1.5 (Vicuna-13B) | 72.3 | | LLaVA-1.5-13B-HD | 86.3 | | SmolVLM-2B | 88.8 |
| InternVL-Chat (Vicuna-13B) | 66.6 | | PaliGemma (3B) | 76.8 | | SmolVLM-2B | 87.5 | | Qwen2-VL-2B | 91.3 |
| **VL-JEPA**SFT **(1.6B)** | 61.5 | | **VL-JEPA**SFT **(1.6B)** | 69.9 | | **VL-JEPA**SFT **(1.6B)** | 85.7 | | **VL-JEPA**SFT **(1.6B)** | 86.3 |

**Results.** Table 1 summarizes the results. In the strict zero-shot setting, VL-JEPABASE achieves higher average accuracy (52.5 vs 44.7) across the 8 classification datasets and higher average recall@1 (63.7 vs 58.1) across the 8 retrieval datasets than the best baseline PE-Core-G. Per-dataset scores show that VL-JEPABASE is particularly strong on *motion-centric* benchmarks (SSv2, EK-100, EgoExo4D, and step recognition on COIN and CrossTask), while relatively weaker on *appearance-centric* benchmarks (Kinetics-400 and task recognition on COIN and CrossTask). This is due to VL-JEPABASE has seen substantially fewer vision-language pairs (only 3.6B in comparison with PE-Core-G's 86B). After supervised finetuning, VL-JEPASFT improves significantly upon VL-JEPABASE since the model has seen in-domain training data. As a single *generalist* model, the performance of VL-JEPASFT is approaching *specialist* models optimized individually for each dataset.

## 4.2 VISUAL QUESTION ANSWERING

**Evaluation Setup.** We evaluate VL-JEPASFT on discriminative VQA tasks. The inference process involves encode candidate answers using the `Y-Encoder` and selecting the answer that minimizes the distance to the predicted embedding (see Fig. 2). We select four benchmarks that prioritize visual perception rather than knowledge and reasoning. We evaluate on GQA (Hudson & Manning, 2019), a dataset for real-world visual reasoning and compositional QA, reporting accuracy on the testdev-balanced split. For TallyQA (Acharya et al., 2019), which targets complex counting, we follow Chen et al. (2022) and report the weighted average accuracy across the "simple" and "complex" splits. Finally, to assess object hallucination, we utilize POPE (Li et al., 2023b) and POPEv2 (Li et al., 2025b). For POPE, we report the average accuracy across the "random", "popular", and "adversarial" settings on MS-COCO.

**Results.** Table 2 compares VL-JEPASFT against established VLM families, including BLIP-2 (Li et al., 2023a), InstructBLIP (Dai et al., 2023), Qwen-VL (Bai et al., 2023), InternVL (Chen et al., 2024d), Llava-1.5 (Vallaeys et al., 2024), SmolVLM (Marafioti et al., 2025), PaLI (Chen et al.,

Table 3: **WORLDPREDICTION-WM benchmark results**. We compare the accuracy between large VLMs, socratic LLMs, and VL-JEPA. VL-JEPA$_{SFT}$ achieves a new SoTA at 65.7%.

| Vision Language Models | | | | | | | | Socratic LLMs (w/ Qwen2.5-VL-72B captions) | | | | | | | | | | VL-JEPA | |
|---|---|---|---|---|---|---|---|---|---|---|---|---|---|---|---|---|---|---|---|
| InternVL2.5 | | | | Qwen2.5-VL | | | | Llama-3.1 | | Llama-4 | | Qwen2.5 | | | GPT-4o | Claude-3.5 | Gemini-2 | BASE | SFT |
| 2B | 4B | 26B | 38B | 3B | 7B | 32B | 72B | 8B | 70B | 109B | 400B | 3B | 7B | 72B | N/A | N/A | N/A | 1.6B | 1.6B |
| 20.0 | 29.8 | 30.2 | 50.3 | 21.6 | 45.5 | 49.0 | 57.0 | 48.7 | 49.8 | 52.7 | 53.6 | 44.0 | 49.1 | 48.5 | 52.0 | 53.3 | 55.6 | 63.9 | **65.7** |

2022), PaliGemma (Beyer et al., 2024), and Video-LLaVA (Lin et al., 2024). VL-JEPA$_{SFT}$ outperforms many of these baselines despite requiring significantly less computational resources–classical VLMs rely on extensively pretrained CLIP backbones combined with multi-stage visual instruction tuning. In comparison, VL-JEPA$_{SFT}$ employs a *unified architecture* and a *single embedding space* to seamlessly handle VQA, classification, and retrieval (Tab. 1).

## 4.3 WORLDPREDICTION-WM

**Evaluation Setup.** We evaluate VL-JEPA on the "world modeling" task in the WORLDPREDICTION (Chen et al., 2025a) benchmark, where the model is provided with two images representing the initial and final world states and must identify, among four candidate video clips, the action that explains the observed transition. To adapt VL-JEPA, we duplicate and concatenate the initial and final state images to extract a *state embedding*, and encode each action candidate into *action embeddings*. The model then selects the candidate whose embedding is closest to the state embedding.

**Results.** Table 3 shows accuracy comparisons. VL-JEPA$_{BASE}$ attains **63.9%** and VL-JEPA$_{SFT}$ attains **65.7%** top-1 accuracy on WORLDPREDICTION-WM, establishing a new state of the art. Our VL-JEPA model not only substantially surpasses existing VLMs of comparable or larger scale but also exceeds the performance of frontier LLMs such as GPT-4o, Claude-3.5-sonnet, and Gemini-2.0.

## 4.4 EMBEDDING PREDICTION VS. TOKEN PREDICTION: A CONTROLLED COMPARISON

**Evaluation Setup.** In this section, we compare VL-JEPA to a token-generative VLM baseline under a strictly aligned training conditions. Both models use the same Perception Encoder (Bolya et al., 2025) (frozen ViT-L-14 with $336^2$ resolution, no tiling, 16 frames per video) for vision inputs. We use the same training iterations with the same effective batch size of 128, same learning rate scheduler on the same pretraining data mixture described above (§3). The only difference is the prediction task: VL-JEPA predicts target embeddings (Duquenne et al., 2023) using a 0.5B predictor, whereas the VLM baseline performs next-token prediction with cross-entropy using a 1B LLM. For VLM, we use the standard training recipe and codebase of PerceptionLM (Cho et al., 2025), aligning frozen vision encoder and text-only LLM `Llama-3.2-1B`. For VL-JEPA, we initialize the predictor from the 8-16 layers of `Llama-3.2-1B`.

We evaluate both models at regular checkpoints throughout training spanning from 500K to 15M samples seen. At each checkpoint, we measure the performance on video captioning and video classification. For video captioning, we report CIDEr scores averaged across YouCook2 (Zhou et al., 2018), MSR-VTT (Xu et al., 2016) and PVD-Bench (Bolya et al., 2025). VL-JEPA decodes the predicted embeddings while VLM generates the tokens directly. For video classification, we report top-5 accuracy averaged across CrossTask-Step, CrossTask-Task (Zhukov et al., 2019) and EgoExo4D (Grauman et al., 2024). For VL-JEPA we choose the candidate with lowest cosine distance to the predicted embedding, while for VLM we pick the class with lowest perplexity.

**Results.** As shown in Fig. 3, both models yield comparable performance after 500K samples seen in both tasks, with respectively 1.23 and 1.35 CIDEr in video captioning and 14.9% and 14.0% top-5 accuracy for VL-JEPA and VLM. After a few iterations, we show that VL-JEPA's performance increase is much sharper compared to VLM, reaching 14.7 CIDEr and 35.3% top-5 accuracy after 5M samples seen. This gap remains constant as training scales at 15M samples with 14.8 CIDEr and 41.0% top-5 accuracy for VL-JEPA, while the VLM baseline yield respectively 7.1 CIDEr and 27.2% top-5 accuracy. This controlled comparison highlights the benefit of predicting embeddings rather than tokens, showing both higher sample efficiency and stronger absolute performance.

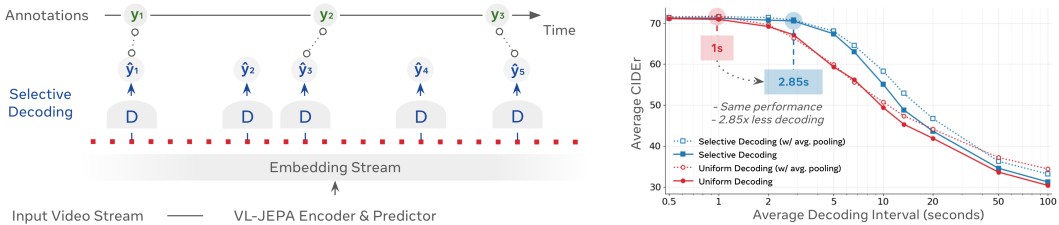

Figure 3: **Comparison of embedding prediction (VL-JEPA) and token prediction (VLM).** We conduct a fair comparison of under strictly aligned training settings (encoder, data, batchsize, etc.). **Left:** Zero-shot video captioning CIDEr score averaged over 3 datasets and zero-shot classification accuracy (top-5) averaged over 3 benchmarks. **Right:** Comparing the trainable parameters and average inference time cost.

Figure 4: **Evaluation of selective decoding. Left:** We compare uniform sampling of decoding points at fixed intervals (red) and embedding-guided selective decoding (blue). Performance is measured by the average CIDEr score between each annotation $y$ and its closest decoded output $\hat{y}$. **Right:** Results on EgoExo4D show that selective decoding achieves a Pareto improvement over uniform sampling: for the same performance level, it requires fewer decoding operations.

We compare the inference cost of the above VL-JEPA and the VLM by pre-loading 64 video frames into memory and repeatedly decoding text 100 times with the same prompt, measuring the average time per sample. As shown in Fig. 3 (right most), both models exhibit comparable latency when generating text. What differentiates our model from classical VLM is the decoupling between the prompt processing ("Query Embedding") and the video encoder ("Encoder + Predictor") from the text generation module ("Decoder"). This allows us to only use the first part of the model to perform retrieval and decode text only when needed (see Section 4.5 below), making our model more scalable for online video inference.

## 4.5 EFFECTIVENESS OF SELECTIVE DECODING

**Evaluation Setup.** We evaluate the effectiveness of VL-JEPA's embedding-guided selective decoding on long-form video streams. To this end, we design a benchmark task where the goal is to recover a temporal sequence of annotations while minimizing the number of text decoding operations, which dominate inference cost. As shown in Fig. 4 (left), decoding is performed only at selected points along the VL-JEPA embedding stream, yielding a sequence of $N$ decoded outputs $[(\hat{t}_1, \hat{y}_1), (\hat{t}_2, \hat{y}_2), \ldots, (\hat{t}_N, \hat{y}_N)]$. Each ground-truth annotation $[(t_1, y_1), (t_2, y_2), \ldots, (t_T, y_T)]$ is then aligned to its nearest decoded output in time (illustrated as $\circ \cdots \circ$ in Fig. 4), and CIDEr is computed between matched pairs. We use the EgoExo4D (Grauman et al., 2024) validation set in procedural activity domains, which consists of 218 videos with an average duration of 6 minutes and about $T = 143$ atomic action annotations per video.

As a baseline, we consider *uniform sampling*, where decoding points are placed at fixed intervals regardless of the underlying video content. Standard streaming VLMs are limited to this strategy, whereas VL-JEPA supports a more effective alternative: *adaptive selection* of decoding points guided by its predicted embeddings. We apply agglomerative clustering with temporal connectivity constraints (Murtagh & Contreras, 2012) to partition the embedding sequence into $N$ segments of high intra-segment monosemanticity (Chen et al., 2024a), measured by variance (*i.e.,* Ward distance). The intuition is that within a semantically coherent segment, decoded outputs are highly similar, so decoding once per segment captures the essential information while greatly reducing overall decoding cost. The midpoint of each segment is then chosen as the decoding point, and de-

Table 4: **Comparison of text-encoders performance**. We report triplet-based accuracy (%) on SugarCrepe++ and VISLA datasets.

| Model | | # Params. (total) | # Params. (text encoder) | Average | SugarCrepe++ (Dumpala et al., 2024a) | | | | | VISLA (Dumpala et al., 2024b) | | |
| | | | | | Replace Attribute | Replace Object | Replace Relation | Swap Attribute | Swap Object | Average | Generic | Spatial |
|---|---|---|---|---|---|---|---|---|---|---|---|---|
| CLIP | ViT-L | 389M | 85M | 44.5 | 56.7 | 83.0 | 42.5 | 27.0 | 13.5 | 34.5 | 37.6 | 31.3 |
| SigLIP2 | ViT-g | 1.9B | 708M | 56.5 | 66.9 | 74.4 | 52.1 | 58.4 | 30.6 | 40.4 | 48.7 | 32.1 |
| PE-Core | ViT-G | 2.3B | 537M | 58.6 | 73.6 | 90.6 | 48.9 | 53.2 | 26.5 | 38.3 | 45.2 | 31.4 |
| VL-JEPA$_{BASE}$ | ViT-L | 1.6B | 300M | 63.9 | 72.2 | 90.1 | 52.2 | 62.9 | 42.0 | 42.9 | 49.8 | 35.9 |
| VL-JEPA$_{SFT}$ | ViT-L | 1.6B | 300M | 58.4 | 68.5 | 90.9 | 47.4 | 55.4 | 29.8 | 39.5 | 44.8 | 34.2 |

coding is performed either from the exact embedding or from the average-pooled embedding within the segment.

**Results.** As shown in Fig. 4 (right), we sweep the average decoding frequency from 2.0 Hz down to 0.01 Hz (*i.e.,* average intervals between consecutive decoding operations from 0.5s to 100s) by adjusting either the stride of uniform sampling or the number of clusters in adaptive selection. Across the entire range, adaptive selection consistently Pareto-dominates uniform sampling. In particular, selective decoding at 0.35 Hz (*i.e.,* $\sim$2.85s interval) matches the performance of uniform decoding at 1 Hz, reducing decoding cost by $\sim$2.85$\times$. We further observe that average pooling provides consistent gains for both strategies, since it provides denoising and stabilization on embeddings prior feeding into the decoder.

### 4.6 EVALUATION OF Y-ENCODER

**Evaluation Setup.** We evaluate whether the JEPA architecture improves the `Y-Encoder` by following the uni-modal text-only (TOT) evaluation setup. We use the hard-negative benchmarks SugarCrepe++ (Dumpala et al., 2024a) and VISLA (Dumpala et al., 2024b). These datasets test sensitivity to semantic and lexical changes in image descriptions. Each dataset contains triplets: two semantically similar descriptions of the same image ($p1$ and $p2$), and one negative description ($n$) created by altering attributes, relations, or objects. We compare `Y-Encoders` from different models by computing the cosine similarity for all description pairs. We check that the similarity between positives $sim(p1, p2)$ is higher than both the similarity between each positive and the negative $sim(p1, n)$ and $sim(p2, n)$. We report accuracy (%) across all samples.

**Results.** Table 4 shows the performance of different models on text hard-negative benchmarks. VL-JEPA$_{BASE}$ achieves a micro average accuracy of 63.9% on SugarCrepe++ and 42.9% on VISLA. This is higher than the best other models: PE-Core scores 58.6% on SugarCrepe++ and SigLIP2 scores 40.4% on VISLA. The finetuned VL-JEPA$_{SFT}$ model also achieves competitive results, with 58.4% on SugarCrepe++ and 39.5% on VISLA. These results indicate that VL-JEPA$_{BASE}$ has a `Y-Encoder` that is more resilient to text hard-negatives.

### 4.7 ABLATION STUDY

**Evaluation Setup.** We study different design choices for VL-JEPA. Here we train all ablation models on the SFT stage data for 10K steps with a batch size of 512 (5M samples seen) and constant learning rate. We report average classification top-1 accuracy of 8 datasets (Tab. 1), average text-to-video retrieval recall@1 of 8 datasets (Tab. 1), and average VQA accuracy of 4 datasets (CLEVR, GQA, TallyQA simple and complex). We report the results in Tab. 5.

**Results. (a) Pretraining.** Dropping the first query-free pretraining stage on image and video captions significantly hurt performance, especially on classification (-21.7) and retrieval (-17.3). **(b) LR Multiplier.** The sweet point of learning rate multiplier to the Y-Encoder is around 0.05 to 0.10. Either faster or slower learning degrades the performance. **(c) Loss Function.** InfoNCE generally give superior performance compared to cosine, L1, and L2 losses, with the only exception being cosine loss outperform InfoNCE on VQA. However, only InfoNCE has the anti-collapse regularization and can be applied with unfrozen Y-Encoder. **(d) Predictor.** In terms of predictor size, more layers yield better performance, especially on VQA performance. We also see that if using the original causal attention instead of updating to bi-direction attention hurt VQA performance (-1.9), since query tokens are appended after visual tokens, and visual tokens are no longer able to attend to

Table 5: **Ablation studies results**. The default setting adopted by VL-JEPA is marked in blue . We calculate ±delta within each group of ablations in comparison with the default setting.

| | Classification (Accuracy) | | Retrieval (Recall@1) | | VQA (Accuracy) | |
|---|---|---|---|---|---|---|
| VL-JEPA$_{SFT}$ | 75.4 | | 63.8 | | 74.2 | |
| *(a) Effectiveness of pretraining stage on caption data* | | | | | | |
| w/ Pretraining | 49.0 | | 47.5 | | 46.1 | |
| w/o Pretraining | 27.3 | (-21.7) | 30.2 | (-17.3) | 42.5 | (-3.6) |
| *(b) Learning rate multiplier for Y-Encoder* | | | | | | |
| *multiplier* = 0.05 | 27.3 | | 30.2 | | 42.5 | |
| *multiplier* = 1.00 | 23.7 | (-3.6) | 28.8 | (-1.4) | 40.7 | (-1.8) |
| *multiplier* = 0.10 | 26.9 | (-0.4) | 30.2 | (-0.0) | 42.9 | (+0.4) |
| *multiplier* = 0.01 | 25.6 | (-1.7) | 27.7 | (-2.5) | 41.0 | (-1.5) |
| *multiplier* = 0.00 | 20.0 | (-7.3) | 25.9 | (-4.3) | 41.4 | (-1.1) |
| *(c) Loss function (with no projection head on top frozen text encoder)* | | | | | | |
| InfoNCE | 23.3 | | 30.3 | | 44.3 | |
| Cosine | 16.5 | (-6.8) | 20.2 | (-10.1) | 46.6 | (+2.3) |
| L1 | 14.8 | (-8.5) | 15.5 | (-14.8) | 41.9 | (-2.4) |
| L2 | 13.5 | (-9.8) | 11.7 | (-18.6) | 43.7 | (-0.6) |

| | Classification (Accuracy) | | Retrieval (Recall@1) | | VQA (Accuracy) | |
|---|---|---|---|---|---|---|
| *(d) Predictor architecture and initialization* | | | | | | |
| Layer 8-16 | 27.3 | | 30.2 | | 42.5 | |
| Layer 0-2 | 24.3 | (-3.0) | 27.8 | (-2.4) | 40.1 | (-2.4) |
| Layer 0-4 | 25.1 | (-2.2) | 28.9 | (-1.3) | 43.6 | (+1.1) |
| Layer 0-8 | 27.2 | (-0.1) | 29.3 | (-0.9) | 43.4 | (+0.9) |
| Layer 0-16 | 27.4 | (+0.1) | 31.0 | (+0.8) | 45.5 | (+3.0) |
| w/o Bi-direction Attention | 26.7 | (-0.6) | 31.2 | (+1.0) | 40.6 | (-1.9) |
| w/o Llama-3 Initialization | 28.1 | (+0.8) | 30.4 | (+0.2) | 40.6 | (-1.9) |
| *(e) Y-Encoder (trainable linear projection on top of frozen text encoder)* | | | | | | |
| EmbeddingGemma-300M | 19.5 | | 24.1 | | 42.5 | |
| Qwen3-Embedding-0.6B | 24.5 | (+5.0) | 24.5 | (+0.4) | 41.5 | (-1.0) |
| Qwen3-Embedding-4B | 27.7 | (+8.2) | 26.6 | (+2.5) | 38.1 | (-4.4) |
| Qwen3-Embedding-8B | 29.6 | (+10.1) | 29.5 | (+5.4) | 41.9 | (-0.6) |
| PE$_{core}$-B (356M) | 29.4 | (+9.9) | 34.5 | (+10.4) | 35.9 | (-6.6) |
| PE$_{core}$-L (356M) | 29.0 | (+9.5) | 34.2 | (+10.1) | 42.9 | (+0.4) |
| PE$_{core}$-G (539M) | 33.9 | (+14.4) | 32.0 | (+7.9) | 41.8 | (-0.7) |

query tokens. Finally, we also see that LLama-3 initialization is beneficial to VQA performance, although vision-language alignment (classification and retrieval) is a bit worse compared to randomly initialized Transformer layers. **(e) Y-Encoder.** We tried different text encoder as the Y-Encoder, and confirmed that VL-JEPA works well with other embedding models than EmbeddingGemma-300M. Generally, larger encoder leads to better performance, with visually aligned text encoders (PE models) has significant advantage in classification and retrieval.

## 5 CONCLUSION

We have present VL-JEPA, a new vision–language model built upon the joint embedding predictive architecture. By shifting supervision from discrete token space to continuous semantic embedding space, VL-JEPA simplifies the learning target, avoids redundant modeling of surface linguistic variability, and enables non-autoregressive prediction. Through controlled experiments, we show that VL-JEPA outperforms generative VLMs trained with cross-entropy loss under matched training data budget, while achieving superior training efficiency and significantly lower inference latency. Beyond generation tasks, the embedding-based design further allows VL-JEPA to handle open-vocabulary classification and cross-modal retrieval within a single unified architecture. Its ability to emit continuous semantic embeddings also makes it particularly well suited for real-time video applications, where selective decoding can improve both responsiveness and efficiency.

### LIMITATIONS

In this work, we demonstrated the advantages of VL-JEPA over standard VLMs, particularly in efficiency, streaming, and video-language tasks. Our goal at this stage, is not to propose a universal alternative to VLMs, as this would require broader evaluation on tasks such as reasoning, tool use, and agentic behaviors where current token generative VLMs excel. Finally, although our results show clear benefits from scaling parameters and dataset size, we did not fully explore this direction, leaving it for future work.

### LLM USAGE

We used large language models (LLMs) solely as writing assistants for this paper. Specifically, they were employed to help rephrase sentences for clarity and readability. No content, ideas, or experimental results were generated by LLMs. The authors take full responsibility for the scientific contributions and all written content.

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

# A  RELATED WORKS

**JEPA Models.** JEPA model learns by predicting the representation of a target input $Y$ from the representation of a context input $X$. Early instantiations include I-JEPA for image encoding (Assran et al., 2023) and V-JEPA for video encoding (Bardes et al., 2023), which demonstrated the effectiveness of this objective over pixel reconstruction approach in their respective modality. Recent JEPA work falls into two categories. One category of work emphasizes better unimodal representation learning (Assran et al., 2023; Bardes et al., 2023; Fei et al., 2023) or cross-modal alignment (Lei et al., 2025; Jose et al., 2025). The other direction targets world modeling, where pretrained encoders are frozen and action-conditioned predictors are trained for conditional prediction of state representations (Zhou et al., 2025; Baldassarre et al., 2025; Assran et al., 2025). This has shown good results but remains limited to narrow domains like mazes or robotic pick-and-place. Our proposed VL-JEPA is the first designed for general-purpose vision–language tasks. It performs conditional latent prediction over vision and text, and preserves efficiency while enabling flexible, multitask architecture.

**Vision Language Models.** Existing vision-language models largely fall into two families: (1) CLIP-style models with a non-predictive joint-embedding architecture (JEA) (Radford et al., 2021; Zhai et al., 2023; Bolya et al., 2025; Liu et al., 2024; Chen et al., 2023) encode images and texts independently into a common latent space, $X_V \mapsto S_V$ and $Y \mapsto S_Y$. By minimizing $\mathcal{L}_{\texttt{CLIP}} = D(S_V, S_Y)$ with a contrastive loss (*e.g.,* InfoNCE), CLIP learns aligned *representations* that support zero-shot classification and vision–language retrieval; (2) Generative VLMs (Liu et al., 2023; Chen et al., 2022; Dai et al., 2023; Alayrac et al., 2022; Chen et al., 2024b; Cho et al., 2025; Beyer et al., 2024) connect a vision encoder (Radford et al., 2021; Fini et al., 2025) with a language model (*e.g.,* LLM). They are typically trained with $\mathcal{L}_{\texttt{VLM}} = D(\hat{Y}, Y)$, *i.e.,* next token prediction with cross-entropy loss, and can learn to handle various vision-text-to-text generation tasks such as visual question answering (VQA).

Our proposed VL-JEPA integrates the architectural advantages and task coverage of both CLIPs and VLMs (Table 6). Since VL-JEPA learns in embedding space, it can leverage web-scale noisy image–text pairs (Jia et al., 2021), yielding strong open-domain features. On the other hand, VL-JEPA supports conditional generation tasks with a readout text decoder. Meanwhile, compared to generative VLMs

|  | CLIP | VLM | VL-JEPA |
|---|---|---|---|
| Generation | ✗ | ✓ | ✓ |
| Retrieval | ✓ | ✗ | ✓ |

Table 6: Task coverage comparison.

that optimize directly in data space, VL-JEPA is more efficient at learning in the latent space. In addition, it is also more efficient for online inference, as it allows naturally selective decoding.

**Efficient Vision Language Models.** The growing size and training cost of VLMs has motivated efforts to improve efficiency. On the training side, strong performance can be achieved by updating only a subset of parameters, such as the vision–language connector (Tsimpoukelli et al., 2021; Alayrac et al., 2022; Vallaeys et al., 2024; Shukor et al., 2023; Koh et al., 2023; Merullo et al., 2022; Dai et al., 2023). At inference, efficiency is pursued through pruning parameters or visual tokens (Cao et al., 2023; Shukor & Cord, 2024; Vasu et al., 2025). For real-time use cases, recent work explores small VLMs (Yao et al., 2024; Marafioti et al., 2025) and heuristics to reduce query frequency in asynchronous inference (Shukor et al., 2025).

**Latent-space Language Modeling.** Current state-of-the-art LLMs are trained to decode and reason in text space using autoregressive generation and chain-of-thought prompting (Wei et al., 2022). Text-space LLMs have rapidly improved and now achieve strong results on a wide range of benchmarks. However, the discrete nature of their reasoning trace may limit both speed and performance in the long term. Several works have explored latent-space LLMs that process or reason in latent space, such as Large Concept Models (Barrault et al., 2024) and COCONUT (Hao et al., 2024). These models focus on unimodal latent-space reasoning. With VL-JEPA, our goal is to align vision and text representations in a shared multi-modal latent space. This approach aims to enable better abstractions and improve both the performance and speed of vision-language models (VLMs). We hope VL-JEPA will serve as a foundation for future work on multi-modal latent space reasoning, including visual chain-of-thought methods (Li et al., 2025a).

