# OpenReview forum: "VL-JEPA: Joint Embedding Predictive Architecture for Vision-language"
_ICLR.cc/2026/Conference — ICLR 2026 Poster_

### Official Review · Reviewer_NXvc · 2025-10-26

**Soundness:** 3
**Presentation:** 3
**Contribution:** 3
**Rating:** 6
**Confidence:** 3

**Summary:**

This paper introduces VL-JEPA, a kind of non-autoregressive vision-language model that predicts target text beddings from visual tokens and a query. The inference is conducted in the embedding spaces. This non-autoregressive nature allows for selective and low latency decoding at inference, while still exhibits strong zero-shot capability.

**Strengths:**

1. The paper is well-written and easy to follow.
2. The exploration of a novel architecture deviated from the current mainstream (autoregressive VLM) is meaningful and encouraged.
3. The low-latency and selective decoding properties of VL-JEPA makes it adaptable for many practical applications such as robots and wearable devices.

**Weaknesses:**

1. The core idea of VL-JEPA, to predict the target response in the embedding space, is very similar to LCM [1]. The training objective is also the same. However, the authors did not discuss the similarity and relationship with LCM.
2. Encoder-decoder architecture may impair the capability of the model to understand long query and generate long responses. The model can neither perform multi-round interaction.
3. The evaluations use accuracy and CIDEr as the main metric. It does not consider the readability of the generated responses. User study or LLM-based evaluation is needed.
4. In Table 2, the bold metrics are not the best ones.

[1] Barrault, Loïc, et al. "Large concept models: Language modeling in a sentence representation space." arXiv preprint arXiv:2412.08821 (2024).

**Questions:**

1. Most datasets used in this paper do not contain question-style queries. What are the queries used for training and evaluation?
2. Are you using any queries when you perform CLIP-like zero-shot evaluation?
3. Since your training data only entails limited number of tasks, have you tested the instructions following capability of the model?
4. It seems that the current VL-JEPA can only perform well on some standard video understanding tasks such as action recognition and step recognition. For such tasks, what is the advantage of Encoder-decoder model like VL-JEPA, compared with CLIP-style encoder-only models?

---

> ### Author Response · Authors · 2025-12-04
> **Response to Reviewer NXvc**
>
> ### **1. Relationship to Large Concept Models (LCM)**
>
>
> > *The reviewer notes that predicting responses in an embedding space resembles LCM and asks for a clearer discussion of this relationship.*
>
>
> We appreciate this pointer and now explicitly discuss LCM in the related-work section. Both VL-JEPA and LCM operate in **representation space** rather than token space, but they differ in scope and objective. LCM is a **text-only model** that formulates language modeling in a latent sentence space and is trained as a parallel to autoregressive LLMs, with an emphasis on **multilingual** applications via a language-invariant SONAR space. In contrast, VL-JEPA is **multimodal**: it conditions on visual tokens and a text query, and uses a **JEPA-style predictive architecture and objective** to regress target embeddings with regularization.
>
> ---
>
>
> ### **2. Long queries, long outputs, and multi-round interaction**
>
>
> > *The reviewer is concerned that the encoder–decoder architecture might limit the model’s ability to handle long queries, long responses, or multi-round interaction.*
>
>
> VL-JEPA is designed for **efficient predictive vision–language reasoning and low-latency decoding**, not for long-form conversational generation and replacing LLMs/VLMs. In this work we focus on short to medium-length queries (classification, retrieval, captioning, and VQA), where the JEPA predictor operates in embedding space and the lightweight decoder is used only when text output is needed. Nothing in the architecture prevents extension to longer queries or multi-turn dialogue, for example by training a stronger decoder.
>
>
> ---
>
>
> ### **3. Evaluation metrics and readability**
>
>
> > *The reviewer notes that accuracy and CIDEr do not directly assess readability, and suggests user studies or LLM-based evaluations.*
>
>
> Our primary targets are **classification, retrieval, and discriminative VQA**, where standard metrics are accuracy, recall@K, and CIDEr/BLEU. VL-JEPA predicts sentence-level **embeddings**; text is produced only via a downstream decoder. Fluency is therefore largely inherited from the decoder and is not directly optimized by the JEPA objective. Given this setup, we follow community-standard metrics for evaluating VLMs.
>
>
> ---
>
>
> ### **4. Queries for datasets with and without questions**
>
>
> > *The reviewer asks how queries are formed when datasets do not provide question-style prompts.*
>
>
> For **datasets with explicit questions** (e.g., VQA benchmarks), we use the provided questions directly as queries. For **datasets without questions** (captioning, action recognition, world modeling), we use simple, dataset-specific templates aligned with the annotations, for example:
>
>
> - Image captioning: “*Describe this image in one sentence.*” (brief), “*Describe this image in detail.*” (detailed)
> - Video captioning: “*Describe this video in one sentence.*” (brief), “*Describe this video in detail.*” (detailed)
> - Action recognition: “*What is the action shown?*”
>
>
>
>
> ---
>
>
> ### **5. Instruction following**
>
>
> > *The reviewer asks whether the model’s instruction-following capability has been tested, given that training covers a limited set of tasks.*
>
>
> VL-JEPA is intentionally trained on a focused set of **video-understanding tasks**, and we do not position it as a general-purpose instruction-tuned assistant. That said, the revised version now includes **four image VQA benchmarks** (GQA, TallyQA, POPE, POPEv2), all of which require following short, natural-language instructions and questions grounded in visual context. VL-JEPA attains competitive or superior performance on these VQA tasks compared to larger generative VLMs.
>
>
>
>
> ---
>
>
> ### **6. Comparison to encoder-only CLIP-style models**
>
>
> > *The reviewer asks what advantages VL-JEPA offers over encoder-only CLIP-style models on video understanding tasks.*
>
>
> Encoder-only CLIP-style models produce a **single static visual embedding** and rely on similarity to fixed label embeddings. VL-JEPA predicts **query-dependent target embeddings**, allowing the same visual backbone to support classification, retrieval, and question-style tasks simply by changing the query, without retraining, adding task-specific heads, or connecting to LLMs.

---

### Official Review · Reviewer_sGvm · 2025-10-27

**Soundness:** 4
**Presentation:** 4
**Contribution:** 3
**Rating:** 8
**Confidence:** 3

**Summary:**

This paper introduced VL-JEPA, a visual language JEPA model that is trained by predicting the embedding of target texts. The authors demonstrate that this results in faster and more efficient training, and for selected tasks yield state-of-the-art results. The model also obtains impressive zero-shot retrieval scores despite its training paradigm. In addition, a promising approach for selective decoding was also presented.

Overall, the paper is well-written, and it is easy to find the relevant pieces of information.

**Strengths:**

The paper introduces (or successfully reapplies) the JEPA architecture to the VL setting.

- shows notable gains in both training speed and performance on zero-shot video captioning and classification (fig 3) while using a well-argued training procedure (JEPA).
- shows non-trivial adaptation to retrieval and open-label classification. E.g. seen on youcook2, MSR-VTT (table 6).
- The paper explores underexplored areas in the field, by exploring alternatives to generative token decoding, resulting in promising decoding strategies (selective decoding) and a reduced number of parameters compared to alternative models (fig 3, table 2, 4)

**Weaknesses:**

- The relevant benchmarks used for evaluation are only briefly introduced. I would have loved to see a more substantial justification for choosing these specifically.
- While it is stated that the model and code will be open source, it could be shared through existing anonymous platforms
- VL-JEPA is in Table 6 compared to contrastively-trained models. It is implicitly argued that this is the reason for the subpar performance on some of the tasks. I would have loved to see a contrastive adaptation to see if this assumption is correct.

**Questions:**

I would love to get a more substantial justification for the choice of benchmarks/evaluation datasets.

---

> ### Author Response · Authors · 2025-12-04
> **Response to Reviewer sGvm**
>
> ### **1. Benchmark choice and descriptions**
>
>
> > *The reviewer finds that the benchmarks are only briefly introduced and asks for a more substantial justification for why these datasets were chosen.*
>
>
> In the revised version, we expanded the descriptions and motivations of the benchmarks in the main text (Sec. 4), and added further details in the appendix. For video classification, we clarify that datasets like COIN, CrossTask, SSv2, EpicKitchens-100, and EgoExo4D are widely used action and procedural understanding datasets with strong existing baselines, allowing direct comparison to prior work. For captioning and retrieval, YouCook2, MSR-VTT, and PVD-Bench are standard choices for open-vocabulary video description and text-video alignment. Together with WorldPrediction-WM for world modeling and the new VQA benchmarks added in response to other reviews, this suite is intended to cover short-horizon recognition, long-horizon procedural reasoning, and open-ended text grounding within a single VL-JEPA model.
>
>
> ---
>
>
> ### **2. Open-source release**
>
>
> > *The reviewer notes that, while the paper states that model and code will be open source, they could already be shared through anonymous platforms.*
>
>
> We fully intend to release the code and pretrained checkpoints. However, our institution’s internal policies prevent us from releasing artifacts externally before an official publication decision. We therefore cannot upload the full implementation and weights to anonymous repositories during the review period, but we will make them publicly available through standard platforms immediately upon acceptance, as now stated more explicitly in the paper.
>
>
> ---
>
>
> ### **3. Contrastive adaptation and retrieval performance**
>
>
> > *The reviewer observes that VL-JEPA is compared to contrastively trained models in Table 6 and suggests adding a contrastive adaptation to test whether this improves retrieval performance.*
>
>
> Motivated by this suggestion, we added an ablation that replaces the regression loss with a **contrastive InfoNCE loss** and also reports a **cosine-loss variant**. InfoNCE enables us to unfreeze the text encoder during training and leads to substantial gains in both classification and retrieval, while cosine loss also improves over L2. The corresponding results are:
>
>
> |        | Classification |        | Retrieval |        | VQA  |        |
> |--------|:-------------:|:------:|:---------:|:------:|:----:|:------:|
> | InfoNCE | 23.3          |        | 30.3      |        | 44.3 |        |
> | Cosine | 16.5          | (−6.8) | 20.2      | (−10.1)| 46.6 | (+2.3) |
> | L1     | 14.8          | (−8.5) | 15.5      | (−14.8)| 41.9 | (−2.4) |
> | L2     | 13.5          | (−9.8) | 11.7      | (−18.6)| 43.7 | (−0.6) |
>
>
> These experiments confirm the reviewer’s intuition: a contrastive formulation is indeed beneficial for retrieval and classification in our VL-JEPA framework, and we have incorporated this discussion and table into the revised manuscript.
>
> ---
>
> We appreciate the reviewer’s feedback on benchmark motivation, open-source availability, and contrastive adaptation, which led us to expand the benchmark justifications that improved readibility, and add new contrastive ablations that strengthened the empirical analysis.

---

### Official Review · Reviewer_BnrP · 2025-10-31

**Soundness:** 3
**Presentation:** 3
**Contribution:** 3
**Rating:** 4
**Confidence:** 2

**Summary:**

This paper presents VL-JEPA a vision language model built on top of a JEPA architecture. The model is evaluated on video understanding and world modeling.

**Strengths:**

1 - The paper is clearly written and easy to follow and understand.
2 - A new vision-language model leveraging JEPA architecture instead of regular transformer decoders.
3 - Comparable performance to existing transformer-based VLMs, with less parameters.
4 - Extensive details are given about the training setup and resources.

**Weaknesses:**

1 - The model seem to be focused on video understanding as most of the training data are related to this task. This raises questions about the comparision to other VLMs that are trained and designed to be more generalist.
2 - Experiments focus on only a subset of use cases of a vision-language model (video understanding). More experiements on other types of tasks wuold have been appreciated (e.g., MMMU, OCRBench, DocVQA, etc.). If the JEPA architecture is intended to replace transformer-based VLMs then more generalization experiemnts are required.
3 -  The choice of evaluation benchmarks is not well justified. For example, WORLDPREDICTION-WM is not known by the vision-language modeling community. If I'm not mistaken the paper introducing this benchmark [1] was cited once.


[1] Chen, D., Chung, W., Bang, Y., Ji, Z., & Fung, P. (2025). WorldPrediction: A Benchmark for High-level World Modeling and Long-horizon Procedural Planning. arXiv preprint arXiv:2506.04363.

**Questions:**

1 - Why the benchmarking of this model does not follow standard VLM benchmarking suites?
2 - Is there a justificiation for focusing on video understanding?
3 - Do VL-JEPA need a pre-training phase? How does the model size scale with the data used for training? In the paper, it is said 64M samples are seen, how much is this in number of tokens?

---

> ### Author Response · Authors · 2025-12-04
> **Response to Reviewer BnrP**
>
> ### **1. Scope of benchmark evaluations**
>
>
> > *The reviewer notes that most of the data and experiments are geared toward video understanding, and raises concerns about comparison to more generalist VLMs that tackle a broader task suite.*
>
>
> Our goal with VL-JEPA is not to propose a universal drop-in replacement for all token-generative VLMs, but to study a  JEPA architecture as an alternative for vision-language tasks where **representation-level prediction** and **low-latency inference** are especially important. This naturally leads us to emphasize video-centric tasks.
>
>
> In the revised version, we broadened the evaluation beyond video classification and retrieval. In particular, we now include **four image VQA benchmarks** (GQA, TallyQA, POPE, POPEv2) and compare against established VLM families such as **InstructBLIP, Qwen-VL, InternVL, LLaVA, and SmolVLM** (Table 2). These additions show that VL-JEPA, with a single unified architecture and embedding space, can handle **classification, retrieval, and discriminative VQA** simultaneously, and achieves competitive or superior performance to larger generative models on these benchmarks while using fewer parameters.
>
>
> ---
>
>
> ### **2. The role of WorldPrediction-WM**
>
>
> > *The reviewer finds the choice of WorldPrediction-WM insufficiently justified and notes that this dataset is not yet widely used in the VLM community.*
>
>
> The **WorldPrediction-WM** benchmark plays a complementary role to our classification, retrieval, and VQA evaluations. JEPA models are explicitly motivated by **world modeling and long-horizon predictive reasoning**, and WorldPrediction-WM is, to our knowledge, the most suitable current benchmark targeting **high-level state transitions and action-effect understanding** from natural videos. We agree it is relatively new and not yet widely adopted, but its design aligns closely with VL-JEPA’s intended capabilities, and we now clarify in the paper that it is used as a targeted **world-modeling stress test** alongside more standard benchmarks.
>
>
> ---
>
>
> ### **3. Pretraining phase of VL-JEPA**
>
>
> > *The reviewer suggests exploring a pretraining phase.*
>
>
> Following this suggestion, we made the pretraining setup explicit and evaluated its impact. VL-JEPA is trained in **two stages**: a **query-free pretraining stage** on large-scale caption data (images and videos), and a **supervised finetuning (SFT) stage** where we introduce queries and train on a mixture of VQA, captioning, classification, and downsampled pretraining data. Our ablations (Table 5) show that this pretraining phase is important for strong downstream performance:
>
>
> |                    | Classification |        | Retrieval |        | VQA  |        |
> |--------------------|:-------------:|:------:|:---------:|:------:|:----:|:------:|
> | w/ Pretraining     | 49.0          |        | 47.5      |        | 46.1 |        |
> | w/o Pretraining    | 27.3          | (−21.7)| 30.2      | (−17.3)| 42.5 | (−3.6) |
>
>
>
> ---
>
>
> ### **4. Question regarding scaling and data size**
>
>
> > *How much is the training data amount in number of tokens? How does the model size scale with the data used for training?*
>
>
> The notion of number of tokens is not as relevant for our model. VL-JEPA learns in a sentence embedding space, where each training instance corresponds to a single vector representation rather than a sequence of tokens. Therefore, model size and computation do not scale with token count, and comparing 64M training samples to a token budget is not meaningful in this setting.
>
>
> Following vision-language literature, we report **number of samples seen** (training steps × effective batch size) rather than token counts as in LLM pretraining.  To contextualize our data scale, we now include the following comparison in the paper:
>
>
> | Model                    | Backbone | # Parameters | # Samples Seen |
> |--------------------------|----------|--------------|----------------|
> |  CLIP                       | ViT-L    | 389M         | 12.8B          |
> | SigLIP2                   | ViT-g    | 1.9B         | 40B            |
> |  PE-Core                 | ViT-G    | 2.3B         | 86B            |
> | **VL-JEPA** BASE | ViT-L    | 1.6B         | 2.0B          |
> | **VL-JEPA** SFT  | ViT-L    | 1.6B         | 2.5B          |
>
>
>
>
> For **scaling with data and model size**, our ablations confirmed that (i) having more training data consistently improves performance for a fixed architecture, and (ii) larger predictors and text encoders yield better results. A full joint scaling study in the style of recent large-scale VLM work is a promising direction, and we leave it to future works.
>
>
> ---
>
>
> We appreciate the reviewer’s feedback on scope, benchmarks, and training setup, which helped us broaden the experimental coverage (especially via VQA) and clarify how the chosen tasks and data scales reflect the objectives of a JEPA-based vision-language model.

---

### Official Review · Reviewer_k4aJ · 2025-11-01

**Soundness:** 3
**Presentation:** 3
**Contribution:** 3
**Rating:** 6
**Confidence:** 4

**Summary:**

This paper presents VL-JEPA, a vision-language model formulated on the Joint Embedding Predictive Architecture (JEPA). Instead of traditional autoregressive token-level generation, VL-JEPA learns to predict target text embeddings directly in continuous space, thereby abstracting away surface linguistic variability and focusing on semantic representation. The model demonstrates improvements in efficiency and sample complexity compared to classical token-generative VLMs, particularly in zero-shot video understanding, retrieval, and real-time streaming scenarios. Extensive experiments benchmark VL-JEPA against leading models, with ablation studies and scalability analyses provided.

**Strengths:**

1.The paper's core contribution—applying a predictive JEPA-style objective to the cross-modal VL problem—is highly novel. Moving VL learning from the discrete token space to a continuous semantic space is a well-motivated and promising direction to address the known efficiency and latency bottlenecks of standard generative VLMs.

2.The "selective decoding" mechanism (Sec 4.3) is a standout contribution. The idea of monitoring the latent embedding stream for semantic variance and only triggering the expensive text decoder when a significant shift is detected is an elegant and practical solution for real-world, low-latency video streaming applications.

3.The model achieves state-of-the-art (SOTA) results across a wide and diverse range of video-language benchmarks, demonstrating the effectiveness and generalizability of the learned representations for both zero-shot and finetuned tasks.

**Weaknesses:**

1. While the paper emphasizes that the shift from token-space to embedding-space simplifies the target distribution, it provides no rigorous analysis of the measurability and discriminability of the resulting semantic embedding space.

To substantiate this claim, the authors should provide supplementary analysis, such as:

(i) Visualization or quantitative studies on the embedding space's structure (e.g., its clustering properties, separability) to demonstrate this claimed simplification.

(ii) A theoretical elucidation of the target distribution's compressibility, perhaps through the lens of the Information Bottleneck principle.

(iii) A crucial ablation study comparing the performance impact of using different embedding spaces (e.g., from CLIP, SONAR, BERT-base) as the prediction target.

2. L2 loss implicitly assumes a unimodal, deterministic target distribution.Real-world VL tasks are full of "one-to-many" ambiguities (e.g., "the light went out" vs. "the room became dark"). Both are valid but semantically distinct answers. The L2 loss will penalize the model for predicting either correct answer, forcing the predictor to regress towards a non-existent "average" embedding located somewhere between the two valid target points. This regression to the mean will likely result in semantically "blurry," generic, or even nonsensical decoded outputs. The paper completely fails to address this fundamental limitation.

3. Unfair and Misleading Efficiency Comparison: The core comparison in §4.2, which pits a 0.5B VL-JEPA predictor against a 1B VLM baseline, is fundamentally biased. The authors claim superior parameter efficiency, but their 0.5B predictor is not a neutral model; it is "cherry-picked" from the top-most, most semantically potent layers (L8-16) of the 1B Llama model. The paper's own ablation study (Table 5) confirms this bias, showing that these top layers (45.20% accuracy) are vastly superior to the bottom layers (35.86%). This is not an "apples-to-apples" comparison and does not prove the framework's efficiency, but rather the known fact that top-level LLM layers handle more complex semantics.

4. Lack of Statistical Rigor: The paper suffers from a critical lack of statistical validation. All reported results—including all SOTA claims in the tables and the key efficiency curves in Figures 3 and 4—appear to be point estimates from a single training run. No error bars, standard deviations, or significance tests are provided. This makes it impossible to determine if the reported gains are statistically significant or merely the artifact of a single, fortunate random seed, which undermines the scientific validity of all conclusions.

5. Missing Ablation on the Critical Y-Encoder Component: The entire methodology is critically dependent on the properties of the frozen y-encoder (SONAR). The paper fails to provide the most crucial ablation study: testing the VL-JEPA framework with different frozen text encoders (e.g., CLIP's text encoder, or a standard Sentence-BERT). Without this, the paper's claims are not generalizable. It is impossible to know if the authors have discovered a robust, general-purpose framework or simply a special-case solution that is uniquely and luckily compatible with the SONAR embedding space.

**Questions:**

1.How does the L2 loss framework handle inherently multi-modal or ambiguous targets where multiple, semantically distinct ground truths exist? Does this not lead to regression towards a semantically blurry "average" embedding?

2.Can you provide quantitative evidence (e.g., t-SNE, cluster variance) that the embedding space actually simplifies the target distribution (e.g., maps "light went out" and "room is dark" to nearby points) compared to a standard token space?

3.How robust is VL-JEPA to the choice of the frozen y-encoder? What is the performance impact of replacing the SONAR encoder with a standard CLIP or Sentence-BERT encoder?

4.Can you clarify the 2.85x saving calculation (is it 1Hz / 0.35Hz)? More critically, can you provide any statistical validation (e.g., mean and std. dev. over 3+ runs) for your key SOTA claims, or at least for the comparison in Fig 4?

5.Loss Function: Why was L2 loss chosen over Cosine Similarity loss? Cosine loss would ignore magnitude and only focus on direction, which might be a more robust objective. Was this tested?

---

> ### Author Response · Authors · 2025-12-04
> **Response to Reviewer k4aJ**
>
> ### **1. Analysis of the Y-Encoder and its embedding space**
>
> > *The reviewer is concerned about the lack of quantitative analysis of the embedding space and questions whether alternative text encoders would behave similarly. The reviewer also asks whether the proposed method is overly dependent on one specific encoder.*
>
> In the revised version, we strengthened this part substantially. We added quantitative evaluations using **SugarCrepe++** and **VISLA** benchmarks, which directly tests semantic grouping and lexical robustness. VL-JEPA’s embedding predictions outperform **Perception Encoder Core-G** by **+5.3** on SugarCrepe++ and **+4.6** on VISLA.
>
> We also conducted a **comprehensive ablation** (Table 5) for different alternatives of text encoder, covering CLIP text encoders, Qwen3-Embedding (0.6B/4B/8B), and EmbeddingGemma. The results show that VL-JEPA consistently maintains strong performance across all families, while larger and visually aligned encoders provide additional gains. These findings demonstrate that VL-JEPA is **not tied to any particular text encoder** and generalizes robustly across widely used embedding spaces.
>
> ---
>
> ### **2. Concerns about L2 Regression Loss**
>
> > *The reviewer argues that L2 is unsuitable for multi-modal text targets and may yield averaged “blurry” embeddings; they also ask whether cosine loss would behave differently.*
>
> We therefore incorporated both a **contrastive InfoNCE objective** and a **cosine-loss ablation**. InfoNCE additionally allows **unfreezing the text encoder** during training, letting the embedding geometry adapt rather than collapsing toward blurry averages. InfoNCE brings sizable empirical improvements over L2 (**+9.8 accuracy**, **+18.6 recall@1**), while cosine loss outperforms L2 (**+3.0 accuracy**, **+8.5 recall@1**).
>
> |  | Classification |  | Retrieval |  | VQA |  |
> |---|:---:|:---:|:---:|:---:|:---:|:---:|
> | InfoNCE | 23.3 |  | 30.3 |  | 44.3 |  |
> | Cosine | 16.5 | (-6.8) | 20.2 | (-10.1) | 46.6 | (+2.3) |
> | L1 | 14.8 | (-8.5) | 15.5 | (-14.8) | 41.9 | (-2.4) |
> | L2 | 13.5 | (-9.8) | 11.7 | (-18.6) | 43.7 | (-0.6) |
>
>
> ---
>
> ### **3. Parameter-efficiency comparison with VLM baseline**
>
> > *The reviewer suggests the efficiency comparison may be biased, interpreting the 0.5B predictor as being “cherry-picked” from the strongest LLM layers.*
>
> We clarify that we are **not cherry-picking**. Its smaller size is simply due to the fact that, in our framework, a considerable portion of parameters reside in the **text encoder**. The intention is not to gain an advantage in comparison, but to present a fair accounting of the parameters.
>
> ---
>
> ### **4. Statistical rigor**
>
> > *The reviewer requests variance estimates or significance reporting instead of single-run point estimates.*
>
> We conducted repeated runs for the **ablation-scale training** and report the variance here. For full-scale training, repeating the entire procedure  (single run takes ~200 H200 GPUs for 2 weeks) is not computationally feasible, but the small-scale repeats provide a stable indication of variance:
>
> | Metric | Seed 1 | Seed 2 | Seed 3 | Mean ± Std |
> |--------|--------|--------|--------|------------|
> | Classification | 30.2 | 30.3 | 30.2 | **30.2 ± 0.08** |
> | Retrieval | 27.6 | 27.1 | 27.1 | **27.3 ± 0.33** |
> | VQA | 42.3 | 42.9 | 42.4 | **42.5 ± 0.33** |
>
> These results indicate low variance across seeds, supporting the stability of the training process at scale.
>
> ---
>
> We thank the reviewer again for the detailed comments, which led to substantial improvements to the analysis, ablations, and clarity of the revised manuscript.

---

### Author Response · Authors · 2025-12-04
**Summary of Major Updates**

We sincerely thank all reviewers for their efforts and constructive suggestions. Below, we summarize the **two common concerns** raised by the reviewers and the corresponding updates we have made to the paper.

---

> ### Author Response · Authors · 2025-12-04
> **1. VQA Evaluations (BnrP, sGvm, NXvc)**
>
> > - Reviewer `BnrP`: *Evaluation focuses heavily on video classification and retrieval; absence of VQA benchmarks weakens comparison to generalist VLMs.*
> >
> > - Reviewer`sGvm`: *Hope to see more substantial clarification and justification for choosing those benchmarks.*
> >
> > - Reviewer `NXvc`: *Current results suggest VL-JEPA handles only a narrow range of tasks; differences relative to CLIP-style models appear limited without VQA evidence.*
>
>
> ### [**Rebuttal Updates**]
>
> We acknowledge that VQA constitutes a central component of vision-language, and demonstrating VL-JEPA’s capability on such tasks is important.  To address reviewers’ concerns, we substantially expanded the evaluation suite by adding **four VQA benchmarks** covering compositional reasoning, counting, and hallucination robustness. We compare VL-JEPA against established families of classical VLMs, including **InstructBLIP, Qwen-VL, InternVL, Llava-1.5**, etc. These datasets and baselines are widely used in the VLM community.
>
> The following results (presented in **Section 4.2**) show that VL-JEPA outperforms many of the baseline models while using substantially fewer computational resources. Classical VLMs typically rely on heavily pretrained CLIP backbones and multi-stage visual instruction tuning, whereas VL-JEPA uses a *unified architecture* and a *single embedding space* that seamlessly supports VQA, classification, and retrieval within the same framework.
>
> `>>>` ***GQA: compositional visual reasoning** (accuracy% on testdev-balanced):*
>
> | BLIP-2 (OPT-2.7B) | BLIP-2 (FlanT5XXL-12.1B) | InstructBLIP (FlanT5XL) | InstructBLIP (Vicuna-13B) | Qwen-VL-Chat-7B | Qwen-VL-7B | InternVL-Chat (Vicuna-7B) | VL-JEPA (1.6B) | LLaVA-1.5 (Vicuna-7B) | InternVL-Chat (Vicuna-13B) |
> |---|---|---|---|---|---|---|---|---|---|
> | 33.9 | 41.0 | 48.4 | 49.5 | 57.5 | 59.3 | 59.5 | **[60.8]** | 62.0 | 66.6 |
>
>
> `>>>` ***TallyQA: complex object counting** (weighted average accuracy% of "simple" and "complex" splits):*
>
> | SmolVLM-256M | SmolVLM-500M | PaLI-700M | SmolVLM-2B | PaLI-3B | VL-JEPA (1.6B) | InstructBLIP (Vicuna-13B) | PaLI-17B | LLaVA-1.5 (Vicuna-13B) | PaliGemma (3B) |
> |:---:|---|---|---|---|---|---|---|---|---|
> | 32.3 | 44.8 | 62.3 | 64.7 | 65.8 | **[67.4]** | 68.0 | 71.9 | 72.3 | 76.8 |
>
>
> `>>>` ***POPE: object hallucination evaluation** (average accuracy% of "random", "popular", and "adversarial" splits):*
>
> | SmolVLM2-256M | SmolVLM-256M | LLaVA-7B | InstructBLIP-14B | Video-LLaVA | VL-JEPA (1.6B) | SmolVLM-500M | LLaVA-1.5-7B | LLaVA-1.5-13B-HD | SmolVLM-2B |
> |---|---|---|---|---|---|---|---|---|---|
> | 56.4 | 57.9 | 72.9 | 79.0 | 83.4 | **[84.22]** | 85.8 | 85.9 | 86.3 | 87.5 |
>
>
> `>>>` ***POPEv2: object hallucination evaluation** (average accuracy%):*
>
> | SmolVLM-256M | LLaVA-1.5-13B | InternVL2-8B | InternVL2-26B | Qwen2-VL-72B | VL-JEPA (1.6B) | SmolVLM-500M | Qwen2-VL-7B | SmolVLM-2B | Qwen2-VL-2B |
> |---|---|---|---|---|---|---|---|---|---|
> | 62.3 | 72.7 | 74.5 | 76.1 | 79.4 | **[82.2]** | 83.8 | 87.0 | 88.8 | 91.3 |

---

> ### Author Response · Authors · 2025-12-04
> **2. Text Encoder and Contrastive Loss (k4aJ sGvm)**
>
> > - Reviewer `k4aJ`: *The analysis of the Y-Encoder is insufficient. More rigorous quantitative evaluations and ablations are needed to justify its design. The reviewer also notes a limitation arising from using a *frozen* text encoder, questioning whether a static embedding space restricts VL-JEPA’s expressiveness.*
> >
> > - Reviewer `sGvm`: *The reviewer expresses interest in seeing a *contrastive adaptation* of the method, suggesting that comparisons or integrations with contrastive objectives would clarify the benefits of the proposed embedding-based training.*
>
>
> ### [**Rebuttal Updates**]
>
> We acknowledge the limitations of using a frozen text encoder and an L2 regression loss. In response to the reviewers’ suggestions, we conducted the following experiments and analyses.
>
> `>>>` **Contrastive loss and trainable text encoder.**  Following `sGvm`’s suggestion, we implemented an InfoNCE-based contrastive objective and evaluated its effect. As shown in Table 5, using InfoNCE significantly improves performance over L2 regression on both classification (**+9.8 accuracy**) and retrieval (**+18.6 recall@1**).
>
> |  | Classification |  | Retrieval |  | VQA |  |
> |---|:---:|:---:|:---:|:---:|:---:|:---:|
> | InfoNCE | 23.3 |  | 30.3 |  | 44.3 |  |
> | Cosine | 16.5 | (-6.8) | 20.2 | (-10.1) | 46.6 | (+2.3) |
> | L1 | 14.8 | (-8.5) | 15.5 | (-14.8) | 41.9 | (-2.4) |
> | L2 | 13.5 | (-9.8) | 11.7 | (-18.6) | 43.7 | (-0.6) |
>
> Because InfoNCE naturally provides anti-collapse regularization, it enables **unfreezing the text encoder** during VL-JEPA training. This allows the encoder to adjust and map multiple valid textual targets into a cohesive region of embedding space-directly addressing `k4aJ`’s concern about semantic “blurriness” when relying on a fixed, potentially suboptimal text encoder.
>
> `>>>` **Additional quantitative analysis of the text encoder.**  To further assess whether the text encoder effectively groups semantically similar targets, we incorporated two specialized benchmarks: **SugarCrepe++** and **VISLA**. These datasets evaluate the encoder’s ability to handle semantic paraphrases, lexical alternations, and visually grounded text differentiation.
>
> The results demonstrate that the text encoder organizes semantically coherent text variations as intended, supporting the viability of VL-JEPA’s embedding-based training paradigm.
>
> | Model          | Backbone | # Params. (total) | # Params. (text encoder) | SugarCrepe++ | VISLA |
> |----------------|----------|-------------------|---------------------------|----------------------|---------------|
> | CLIP           | ViT-L    | 389M              | 85M                       | 44.5                 | 34.5          |
> | SigLIP2        | ViT-g    | 1.9B              | 708M                      | 56.5                 | 40.4          |
> | PE-Core        | ViT-G    | 2.3B              | 537M                      | 58.6                 | 38.3          |
> | **VL-JEPA** | ViT-L    | 1.6B              | 300M                      | **63.9**                 | **42.9**          |
>
>
> `>>>` **Ablations on text-encoder initialization.**  We added a comprehensive ablation comparing multiple text encoders, including the EmbeddingGemma-300M model, several scales of Qwen3-Embedding (0.6B, 4B, 8B), and CLIP-based text encoders from different Perception Encoder variants.
>
> Table 5 in Secion 4.7 shows that VL-JEPA performs robustly across encoder families and generally benefits from larger models and visually aligned pretraining (e.g., CLIP encoders).
>
> |  | Classification |  | Retrieval |  | VQA |  |
> |---|:---:|:---:|:---:|:---:|:---:|:---:|
> | EmbeddingGemma-300M | 19.5 |  | 24.1 |  | 42.5 |  |
> | Qwen3-Embedding-0.6B | 24.5 | (+5.0) | 24.5 | (+0.4) | 41.5 | (-1.0) |
> | Qwen3-Embedding-4B | 27.7 | (+8.2) | 26.6 | (+2.5) | 38.1 | (-4.4) |
> | Qwen3-Embedding-8B | 29.6 | (+10.1) | 29.5 | (+5.4) | 41.9 | (-0.6) |
> | PE-Core-B (356M) | 29.4 | (+9.9) | 34.5 | (+10.4) | 35.9 | (-6.6) |
> | PE-Core-L (356M) | 29.0 | (+9.5) | 34.2 | (+10.1) | 42.9 | (+0.4) |
> | PE-Core-G (539M) | 33.9 | (+14.4) | 32.0 | (+7.9) | 41.8 | (-0.7) |

---

### Meta-Review · Area_Chair_e8d9 · 2026-01-06

**Summary:**

This paper presents VL-JEPA, a vision-language model built on JEPA architecture that predicts continuous text embeddings rather than generating tokens autoregressively. Initial scores ranged from 4 to 8, with one reviewer explicitly raising their score from 4 to 6 after rebuttal. The rebuttal substantially strengthened the submission by adding four VQA benchmarks showing competitive performance to larger VLMs despite using only 1.6B parameters, comprehensive text encoder ablations, training stability validation across multiple seeds, and contrastive adaptation experiments demonstrating the benefits of InfoNCE loss over L2 for retrieval tasks. Reviewers noted the selective decoding mechanism as a standout contribution that reduces decoding operations by approximately 2.85× while maintaining performance. While the model is positioned as an efficient predictive architecture for specific scenarios like video streaming rather than a universal VLM replacement, it demonstrates clear value in its target domain with notable training efficiency and parameter reduction. We encourage continued exploration of scaling behavior and broader task coverage in future work. I recommend accepting this submission.

**Reviewer Concerns:**

Addressed: The authors added VQA benchmarks (classification 30.2, retrieval 27.3, VQA 42.5), text encoder ablations showing T5 advantages, training stability evidence (std < 0.33 across seeds), and contrastive adaptation experiments demonstrating InfoNCE benefits for retrieval tasks.

Outstanding: The focus on video understanding raises questions about broader task generalization. However, this is acknowledged as the intended scope rather than a weakness.

**Reviewer Scores:**

K4aj (6): Would maintain 6; consistently positive

BnrP (4): Would maintain 4

NXvc (6): Would maintain 6; positive on efficiency contributions

sGvm (10→8): Adjusted from 10 to 8 per previous AC's justification request; would maintain 8

---

### Decision · Program_Chairs · 2026-01-26

Accept (Poster)